# DprE2 is a molecular target of the anti-tubercular nitroimidazole compounds pretomanid and delamanid

Katherine A. Abrahams [1,3], Sarah M. Batt [1,3], Sudagar S. Gurcha[1], Natacha Veerapen[1], Ghader Bashiri [2] & Gurdyal S. Besra [1] ✉

*Mycobacterium tuberculosis* is one of the global leading causes of death due to a single infectious agent. Pretomanid and delamanid are new antitubercular agents that have progressed through the drug discovery pipeline. These compounds are bicyclic nitroimidazoles that act as pro-drugs, requiring activation by a mycobacterial enzyme; however, the precise mechanisms of action of the active metabolite(s) are unclear. Here, we identify a molecular target of activated pretomanid and delamanid: the DprE2 subunit of decaprenylphosphoribose-2'-epimerase, an enzyme required for the synthesis of cell wall arabinogalactan. We also provide evidence for an NAD-adduct as the active metabolite of pretomanid. Our results highlight DprE2 as a potential anti-mycobacterial target and provide a foundation for future exploration into the active metabolites and clinical development of pretomanid and delamanid.

*Mycobacterium tuberculosis* (*Mtb*), the causative agent of tuberculosis (TB), has been ranked as the leading cause of death from a single infectious agent since 2014. Though superseded more recently by COVID-19, which has also impacted case diagnostics and reporting, in 2021 an estimated 10.6 million people developed active TB disease with 1.6 million associated deaths, exemplifying TB's global burden[1,2]. Despite a gradual reduction in mortality rate over recent years, multi-drug resistant (MDR) and extensively drug resistant (XDR) strains of *Mtb* continue to threaten the efficacy of the current treatment strategy, demanding the urgent introduction of new chemotherapeutic agents and drug regimens. Presently, there are 26 anti-tubercular drugs in Phase I, II and III clinical trials and combination programs in pursuit of the effective treatment of drug-susceptible, MDR and latent forms of TB. These drugs are comprised of 17 new compounds, 6 repurposed drugs and 3 that have obtained regulatory approval and are used in a select population of MDR-TB patients[2]. Two of these approved compounds are the bicyclic 4-nitroimidazoles, pretomanid (PA-824; Global Alliance for TB Drug Discovery) and delamanid (OPC-67683, Deltyba; Otsuka Pharmaceutical Company) (Fig. 1a), which have completed Phase III clinical trials in combination therapy with positive outcomes[3,4]. Consequently, the United States Food and Drug Administration have approved the use of pretomanid in combination with linezolid and bedaquiline in the treatment of specific TB infections[5]. In addition to their potent activity against MDR clinical isolates, pretomanid and delamanid are active against non-replicating TB[6,7], further expanding their therapeutic potential.

Despite the use of pretomanid and delamanid in clinical trials for over a decade, their precise molecular target(s) have remained elusive. It has long been established that both pretomanid and delamanid are pro-drugs, requiring activation by the deazaflavin $F_{420}$-dependent nitroreductase, Ddn[8] (Fig. 1b), expressed solely in the mycobacterial cells not the host[9]. Clinical and laboratory-generated resistant isolates exhibit mutations in either *Ddn* or one of five other genes, all of which are essential in pro-drug activation: *fbiA, fbiB, fbiC, fbiD* or *fgd1*[10–12]. FbiA, FbiB and FbiC are involved in the synthesis of $F_{420}$ and Fgd1 is a $F_{420}$-dependent glucose-6-phosphate dehydrogenase that regenerates the reduced form of the $F_{420}$ co-factor for further cycles with Ddn[10] (Fig. 1b). Mutations in the molecular target have not been documented, and this has prevented advances in target validation that continue to be explored.

[1]Institute of Microbiology and Infection, School of Biosciences, University of Birmingham, Edgbaston, Birmingham B15 2TT, UK. [2]Laboratory of Molecular and Microbial Biochemistry, School of Biological Sciences, University of Auckland, 3A Symonds Street, Auckland 1010, New Zealand. [3]These authors contributed equally: Katherine A. Abrahams, Sarah M. Batt. ✉e-mail: g.besra@bham.ac.uk

Fig. 1 | Structures of pretomanid and delamanid with the mechanism of activation and target inhibition pathway. a Structures of pretomanid and delamanid. The common nitro group is highlighted in red. b Pro-drug activation mechanism with $F_{420}$ redox cycling. Fgd1 catalyzes the oxidation of glucose-6-phosphate to 6-phosphogluconolactone, generating $F_{420}$-$H_2$. This co-enzyme is then oxidized by Ddn in the reduction of the pro-drugs pretomanid or delamanid into active forms, regenerating $F_{420}$. DprE1 catalyzes the oxidation of DPR (decaprenylphosphoryl-D-ribose) into a keto-intermediate, DPX (decaprenylphosphoryl-2-ketoribose), which is subsequently reduced by DprE2, generating DPA (decaprenylphosphoryl-D-arabinose). The activated pro-drugs inhibit the DprE2-catalyzed reduction of DPX. The R group represents the decaprenyl ($C_{50}$) moiety.

Target assignment is valuable in the progression of medicinal chemistry efforts to improve the efficacy, toxicity and bioavailability potential of compounds for clinical use. The current evidence indicates pretomanid and delamanid have a multifaceted mechanism of action, dependent on aerobic and anaerobic conditions, although the active metabolite(s) has yet to be confirmed. During aerobic growth, treatment with pretomanid or delamanid has been shown to affect the mycobacterial cell wall[13], with a reduction in mycolate intermediates pointing towards mycolic acid synthesis as the probable target[6,7]. During the non-replicating phase, cell wall remodeling is limited[14], and it has been proposed that the oxygen-limiting conditions could be more of a factor in the observed respiratory poisoning caused by both drugs. This is thought to occur through the release of reactive nitrogen species (such as nitric oxide (NO) during pro-drug activation), which interferes with electron flow and ATP homeostasis[8,13,15]. However, recent evidence suggests that the presence of an NO scavenger does not impact the antimycobacterial effect of delamanid under hypoxic, non-replicating conditions, suggesting an alternative mechanism is possible[9].

Upon reduction by Ddn, pretomanid and delamanid are converted into many different metabolites, some of which are likely to be unstable and short-lived[8,9]. Synthesis of the more stable metabolites identified has shown that none are active against *Mtb* and that an unstable derivative could represent the active form of the drugs[8]. Efforts to identify the active metabolite have also resulted in the discovery of NAD-adducts in both pretomanid and delamanid-treated *Mtb*[9,16]. However, the antimycobacterial activity of these NAD-adducts has not been observed, possibly owing to low membrane permeability for in vitro studies[9].

In this work, we have identified the essential enzyme, decaprenylphosphoryl-2-keto-β-D-erythro-pentose reductase (DprE2)[17], required for the synthesis of precursors to the cell wall component arabinogalactan, as a molecular target of pretomanid and delamanid (Fig. 1). We also establish the requirement of NAD(H) in the activation of pretomanid, providing further evidence to support the NAD-adduct as the active metabolite and suggesting NADH is the natural cofactor of DprE2. We anticipate that this expansion of knowledge will aid future medicinal chemistry optimizations and direct treatment combinations of these promising anti-tubercular compounds.

## Results

In our continuous pursuit to identify new chemical scaffolds and drug targets to aid in the TB drug discovery pipeline, we endeavored to find inhibitors of unexploited targets within mycobacterial cell wall biosynthesis. The essential decaprenylphosphoribose-2'-epimerase, composed of subunits DprE1 and DprE2, is responsible for the synthesis of the lipid-linked D-arabinose donor, decaprenylphosphoryl-D-arabinose (DPA) (Fig. 1b), which is a precursor to a major cell wall constituent, arabinogalactan[18–21]. DprE1 (decaprenylphosphoryl-β-D-ribose oxidase) is a validated drug target;[22,23] the benzothiazinone (BTZ) compounds, BTZ043 and PBTZ169, targeting DprE1, are currently progressing through clinical trials[24]. Owing to its druggability, DprE1 has been extensively screened for new inhibitory scaffolds. On the contrary, despite its suitability as a drug target, inhibitors of DprE2 have not been

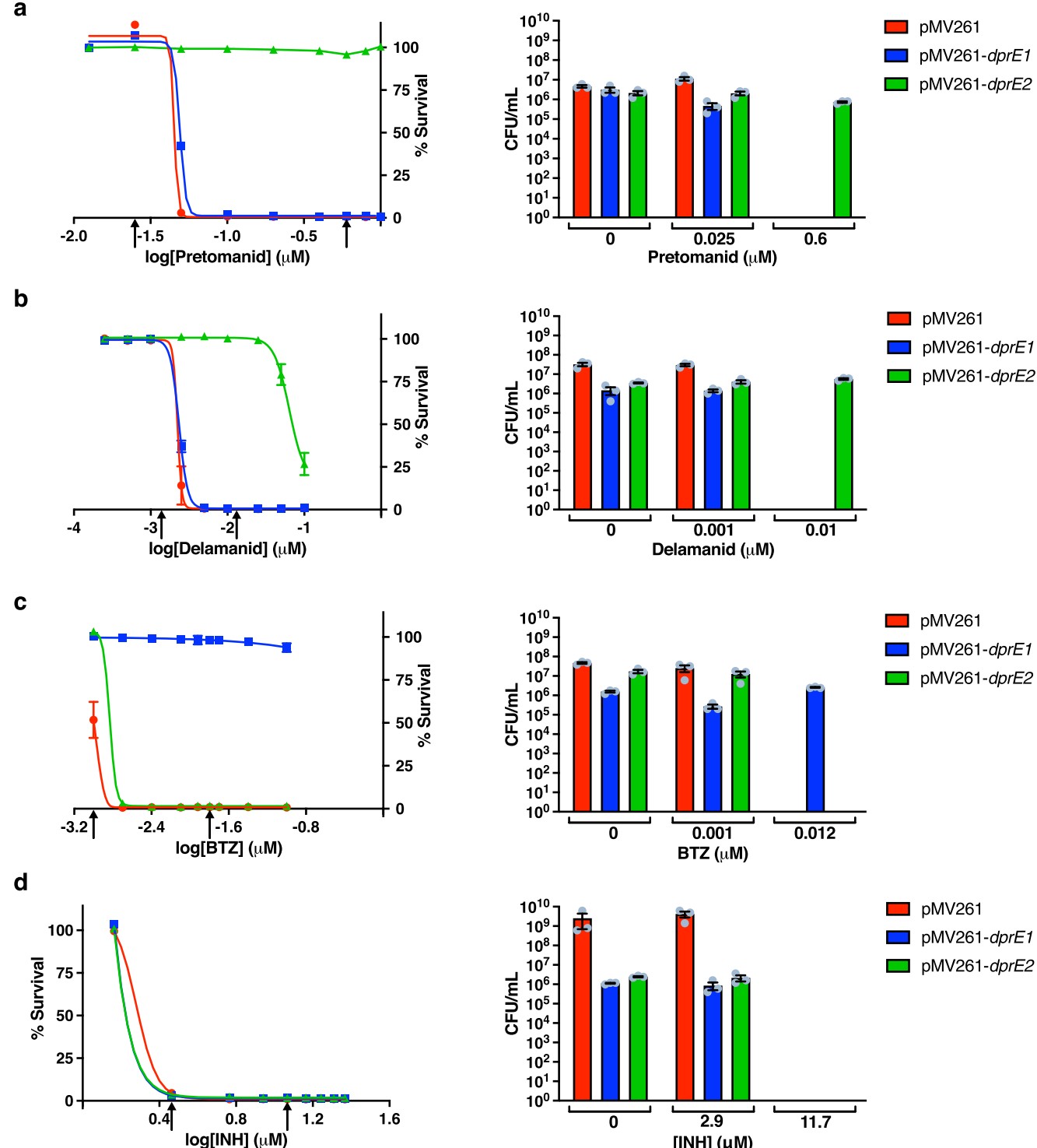

**Fig. 2 | Over-expression of Mt-DprE2 increases resistance to pretomanid and delamanid.** The MICs of **a** pretomanid and **b** delamanid were established using *M. bovis* BCG strains containing the constitutive expression vectors pMV261 (red), pMV261-*Mt-dprE1* (blue) and pMV261-*Mt-dprE2* (green). **c** BTZ was used as a positive control for DprE1 inhibition; **d** isoniazid was used as a negative control for DprE1 and DprE2 inhibition. Cell viability was determined using the MABA and CFU enumeration. Arrows on the MABA graphs indicate the concentration of drug selected for CFU enumeration. Data were plotted and fitted using Prism GraphPad showing the percentage survival (MABA) or CFU/mL (individual data points shown as gray circles), based on the mean of triplicate data (*n* = 3) from independent experiments with bars representing the standard error. Source data are provided as a Source Data file.

described. Therefore, by using a whole cell phenotypic target-based screen, we aimed to identify new compound classes that specifically inhibited DprE2. From this screen, we identified a number of compound hits that were progressed to target validation by whole genome sequencing of spontaneous resistant mutants[25]. The data revealed

mutations in the genes *fgd1* and *fbiC*, common to the mutations found in pretomanid and delamanid resistant isolates. From this, we envisaged that the compounds share the same pro-drug activation pathway and therefore could potentially have the same target. Thus, we investigated whether pretomanid and delamanid did in fact target DprE2.

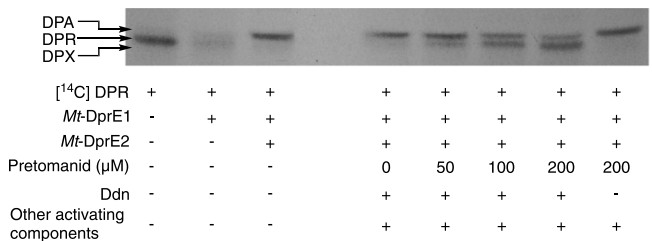

| | | | | | | | | |
|---|---|---|---|---|---|---|---|---|
| [14C] DPR | + | + | + | + | + | + | + | + |
| *Mt*-DprE1 | - | + | + | + | + | + | + | + |
| *Mt*-DprE2 | - | - | + | + | + | + | + | + |
| Pretomanid (µM) | - | - | - | 0 | 50 | 100 | 200 | 200 |
| Ddn | - | - | - | + | + | + | + | - |
| Other activating components | - | - | - | + | + | + | + | + |

**Fig. 3 | Inhibition of radiolabeled [14C]-DPR epimerization to [14C]-DPA by Ddn-activated pretomanid.** The in vitro *Mt*-DprE1-DprE2-dependent epimerization of [14C]-DPR to [14C]-DPA and dose-dependent inhibition by Ddn-activated pretomanid was analyzed by autoradiography TLC (2000 cpm; developed in $CHCl_3$:$CH_3OH$:$NH_4OH$:$H_2O$ (65:25:0.5:3.6, (v/v)); exposed to Kodak X-Omat film for 14 days). The reaction components of each lane are summarized. Pretomanid pro-drug was activated overnight at room temperature in the presence of *Mt*-DprE1/*Mt*-DprE1-DprE2, 200 µM NADPH and activating components (plus/minus Ddn). DPR, decaprenylphosphoryl-D-ribose; DPX, decaprenylphosphoryl-2-ketoribose; DPA, decaprenylphosphoryl-D-arabinose. Source data of 3 TLCs are provided as a Source Data file.

## Over-expression of DprE2 increases resistance to pretomanid and delamanid

Initially, the impact on the minimal inhibitory concentration (MIC) of pretomanid and delamanid was determined using *Mycobacterium bovis* BCG strains over-expressing *Mtb* DprE1 (*Mt*-DprE1) or *Mtb* DprE2 (*Mt*-DprE2), with an empty vector control (Fig. 2). The mycobacterial expression vectors, pMV261 harboring *Mtb* *dprE1* or *dprE2*, were electroporated into *M. bovis* BCG and the Microplate Alamar Blue Assay (MABA) and colony forming unit (CFU) enumeration was used to establish cell viability over a range of drug concentrations. The MABA results clearly demonstrate that, upon over-expression of *Mt*-DprE2 compared to the empty vector control, there is an increase in resistance greater than 20-fold to both pretomanid (MIC 0.05 µM to >1 µM) and delamanid (MIC 0.005 µM to >0.1 µM). No significant change in inhibition was observed between the over-expression of *Mt*-DprE1 and the empty vector (MIC pretomanid, 0.1 µM; delamanid, 0.005 µM). Positive and negative control studies were performed using BTZ (to specifically target DprE1) and isoniazid, respectively. Resistance to BTZ was only observed with the DprE1 over-expressor strain, and no increase in resistance against isoniazid was observed with either *Mt*-DprE1 or *Mt*-DprE2 over-expressor strains. An *Mtb inhA* over-expressor strain was used to show specificity of pretamonid and delamanid for the inhibition of DprE2 and not other ubiquitous NAD(P)H-dependent enzymes (Supplementary Fig. 1). CFU enumeration confirmed the interpretation of the MABA results.

## Ddn-activated pretomanid inhibits DPR epimerization to DPA

To confirm the specific inhibition of DPA synthesis by activated pre-tomanid, an in vitro assay with thin layer chromatography (TLC) analysis was used to analyze the DprE1-DprE2-dependent epimerization of [14C]-DPR (decaprenylphosphoryl-D-ribose) to [14C]-DPA (decaprenylphosphoryl-D-arabinose) (Fig. 1) using recombinant enzymes (either *Mt*-DprE1 or the *Mt*-DprE1-DprE2 complex). Delamanid was omitted from in vitro kinetic assays due to its insolubility in the buffers used. Figure 3 shows the *Mt*-DprE1 enzymatic oxidation of [14C]-DPR (lane 1) to [14C]-DPX (decaprenylphosphoryl-2-ketoribose) (lane 2). This intermediate is rather unstable and migrates as a diffuse band. When coupled directly with *Mt*-DprE2, [14C]-DPX is reduced in situ to [14C]-DPA (lane 3). A dose-dependent inhibition of [14C]-DPA synthesis was observed with increasing concentrations of Ddn-activated pretomanid (0-200 µM; lanes 4–7). In the presence of 200 µM pretomanid (non-Ddn-activated), there was no observable inhibition of [14C]-DPA synthesis (lane 8). These results signify that Ddn-activated pretomanid inhibits the *Mt*-DprE1-DprE2 dependent epimerization of DPR. This is

further corroborated by the analysis of mycobacterial cell wall extracts (arabinogalactan) after [14C]-labeling and treatment with pretomanid, delamanid or BTZ: the relative abundance of arabinose in comparison to galactose decreases with increasing drug concentrations (Supplementary Fig. 2), suggesting DPA synthesis has been negatively impacted. Interestingly, the inhibition of DPR epimerization by Ddn-activated pretomanid did not cause a discernible accumulation of DPX, which would be concomitant with the inhibition of *Mt*-DprE2 alone. The absence of DPX could provide an insight into the mechanism of inhibition and was further investigated by spectrophotometric kinetic analysis of *Mt*-DprE1 and *Mt*-DprE2 activity.

## Spectrophotometric assays for DprE2 inhibition

To further confirm DprE2 as the target of Ddn-activated pretomanid, as well as to gain an insight into the mechanism of inhibition, in vitro assays were used to analyze *Mt*-DprE1 and *Mt*-DprE2 activity. Both assays use the soluble DPR analog, geranylgeranylphosphoryl-D-ribose (GGPR), where DprE1 converts GGPR to the DprE2 substrate, geranylgeranylphosphoryl-2-ketoribose (GGPX); DprE1 activity is therefore a pre-requisite to DprE2 activity. The spectrophotometric assay for DprE1 activity measures resazurin reduction, by the DprE1 $FADH_2$ cofactor, to fluorescent resorufin (excitation 530 nm, emission at 590 nm)[26], whilst the DprE2 assay follows the oxidation of the DprE2 NADPH cofactor (excitation at 340 nm, emission at 445 nm)[25] (Fig. 4a).

## The active form of pretomanid requires NADH and demonstrates tight binding to DprE2

To investigate the requirements for the activation of pretomanid, different combinations of the suggested natural cofactors of DprE2, NADH and NADPH[16], along with the oxidized forms, $NAD^+$ and $NADP^+$ (Fig. 4bi), were incubated with the pro-drug in the presence and absence of the *Mt*-DprE1-DprE2 complex during activation with Ddn. Subsequent analysis of DprE2 activity showed that in the absence of *Mt*-DprE1-DprE2 during activation, NADH or $NAD^+$ was required to convert pretomanid to an active form that could inhibit DprE2 activity. Neither NADPH nor $NADP^+$ showed this ability. However, if *Mt*-DprE1-DprE2 was present during Ddn-activation, the addition of a nicotinamide cofactor was not required for the pretomanid to inhibit DprE2 activity. This was also demonstrated with the radioactive DPR epimerization assays, where NAD(H) was not included. This led us to conclude that the active metabolite of pretomanid is not short-lived as previously thought[8], but instead metabolized to inactive components in the absence of NAD(H). The requirement for *Mt*-DprE1-DprE2 during the formation of an active metabolite, in the absence of additional NAD(H), suggests that the complex is providing an endogenous source of the cofactor and that NADH is therefore the natural cofactor for DprE2. It should be noted that we continued monitoring NADPH oxidation for DprE2 activity as Fgd1 was found to oxidize NADH in our assay (Supplementary Fig. 3).

To assess the stability of the enzyme-inhibitor complex, the inhibited *Mt*-DprE1-DprE2 complex (3 h incubation during pretomanid activation), was dialyzed overnight to dilute out unbound inhibitor molecules and derivatives. DprE2 activity (monitoring NADPH oxidation) of the inhibited *Mt*-DprE1-DprE2 complex (plus Ddn) was compared pre- and post-dialysis with the uninhibited *Mt*-DprE1-DprE2 complex (minus Ddn). There was no discernible effect of the dialysis on the activity of either the inhibited or uninhibited *Mt*-DprE1-DprE2 complex (Fig. 4bii); the inhibited complex was still inhibited post-dialysis with a curve similar to that of the negative control (spontaneous NADPH oxidation only) and the uninhibited complex showed activity comparable to the pre-dialyzed sample. The inability of the inhibited DprE2 to regain activity post-dialysis suggests irreversible inhibition, whereby the inhibitor has bound extremely tightly, or the chemical composition or tertiary structure of the enzyme has been permanently altered.

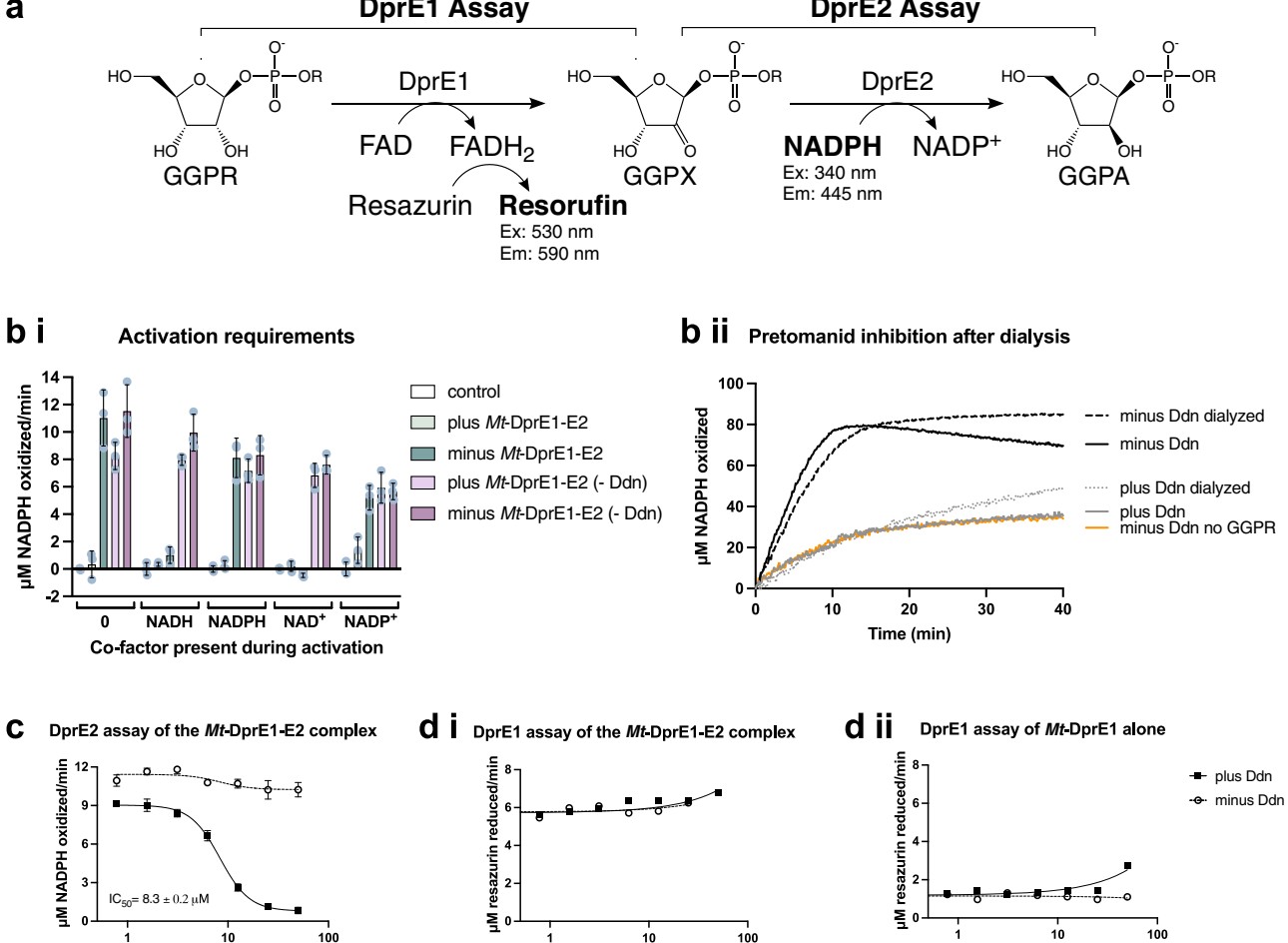

**Fig. 4 | Analysis of pretomanid activation and inhibition of DprE2. a** Schematic representation of the DprE1 and DprE2 assays. DprE1 activity was measured through a fluorescence assay that measured the reduction of resazurin to fluorescent resorufin as DprE1 oxidized GGPR to the keto-intermediate GGPX, using the cofactor FAD. DprE2 activity was monitored through a change in fluorescence due to the oxidation of the DprE2 cofactor, NADPH, as the intermediate GGPX was reduced to the GGPR product. GGPR, geranylgeranylphosphoryl-D-ribose; GGPX, geranylgeranylphosphoryl-2-ketoribose; GGPA, geranylgeranylphosphoryl-D-arabinose. **b** (i) NAD(H) is required for the activation of pretomanid. DprE2 activity was analyzed following pretomanid activation for 3 h at 30 °C in the presence of different combinations of the *Mt*-DprE1-DprE2 complex and the co-factors NADH, NADPH, NAD⁺ and NADP⁺. The controls (reaction mixes minus GGPR) were subtracted to remove background NADPH oxidation. Reactions were performed in triplicate (*n* = 3) (individual data points of replicates shown as gray circles) and mean initial rates (μM NADPH oxidized/min) with the standard error were plotted using Prism GraphPad. **b** (ii) Measuring the stability of the enzyme-inhibitor complex by dialysis. Raw data showing the DprE2 activity (measuring NADPH oxidation) of pretomanid inhibited and uninhibited *Mt*-DprE2, pre- and post-dialysis. Activity assays were performed in triplicate (*n* = 3), with a single representative data set shown. The control (orange; minus Ddn and GGPR) shows spontaneous NADPH oxidation. **c** DprE2 assay monitoring DprE2 activity in the *Mt*-DprE1-DprE2 complex, **d** DprE1 assay monitoring DprE1 activity in (i) *Mt*-DprE1-DprE2 complex and (ii) *Mt*-DprE1 alone, in the presence of increasing concentrations of Ddn-activated and non-activated pretomanid (activated for 1.5 h at 30 °C before adding to *Mt*-DprE1-DprE2/ *Mt*-DprE1). Assays in c and d were performed in triplicate (*n* = 3) and graphs were plotted with [inhibitor] (log₁₀ scale μM) vs. response and IC₅₀ (half maximal inhibitory concentration) calculated fitting a four-parameter dose response curve, using Prism GraphPad. Source data are provided as a Source Data file. *Mt*-DprE1-E2, *Mt*-DprE1-DprE2 complex.

## Kinetics of DprE2 inhibition by pretomanid

Having established the parameters of pro-drug activation, the kinetics of activated pretomanid inhibition were investigated. The DprE2 assay was used to analyze DprE2 activity as part of the *Mt*-DprE1-DprE2 complex; we were unable to purify active DprE2 in the absence of DprE1, and DprE1 activity is a pre-requisite to DprE2 activity. The DprE1 assay was used to determine any impact of the activated inhibitor on the primary step of GGPR epimerization. In both assays, increasing concentrations of pretomanid (0-50 μM) were incubated for 1.5 h at 30 °C with the NAD⁺ and all of the components required for pro-drug activation. Equivalent reactions, omitting Ddn, were also prepared, preventing pro-drug conversion to an active molecule. In the presence of the pretomanid pro-drug (not activated), *Mt*-DprE2 activity was evident. Whereas upon activation of the pretomanid by Ddn, there was a dose-dependent reduction in *Mt*-DprE2 activity, with a calculated half

maximal inhibitory concentration (IC₅₀) value of 8.3 ± 0.2 μM (Fig. 4c; raw data, Supplementary Fig. 4a) (± SEM). Next, the DprE1-catalyzed resazurin reduction was used to evaluate activated pretomanid inhibition of DprE1 activity in the *Mt*-DprE1-DprE2 complex, along with any inhibition of *Mt*-DprE1 alone (Fig. 4d; raw data, Supplementary Fig. 4b and c). There was no inhibition of DprE1 activity with either *Mt*-DprE1 alone or the *Mt*-DprE1-DprE2 complex. This correlates with the over-expression data, whereby increasing cellular levels of DprE1 had no effect on the MIC of pretomanid. The DprE1 inhibitor, BTZ (Supplementary Fig. 5), demonstrated inhibition of the DprE1 assay for both *Mt*-DprE1 and *Mt*-DprE1-DprE2 (IC₅₀ values of 11.8 ± 0.7 μM and 24.0 ± 1.2 μM, respectively) (± SEM).

To obviate the effect of substrate concentration and study the mode of inhibition of DprE2 by pretomanid, the $K_i$ (inhibitor constant) was determined by varying the concentration of both GGPR and

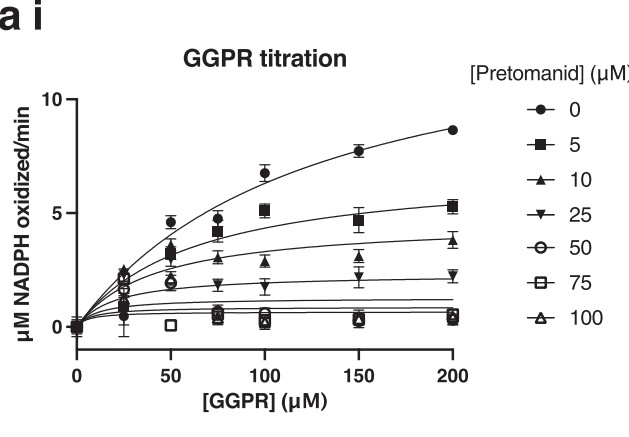

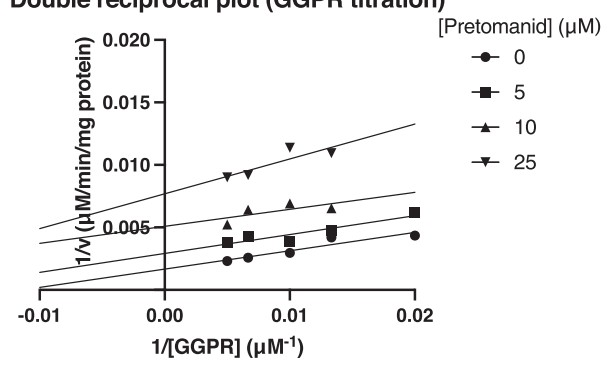

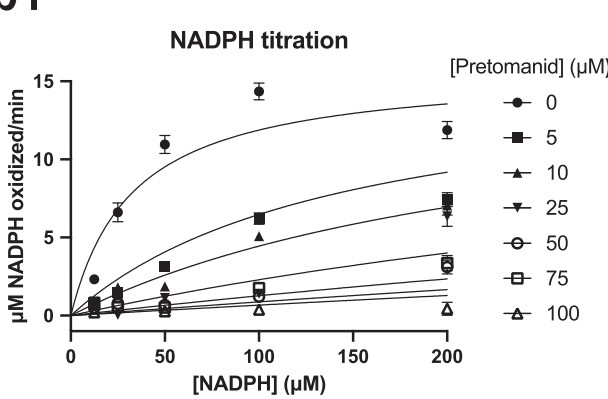

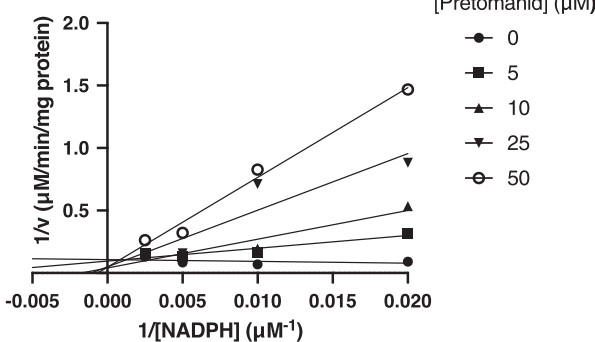

**Fig. 5 | $K_i$ determination of pretomanid for the DprE2 assay.** Pretomanid inhibition was performed with a titration of the inhibitor and the two substrates, **a** GGPR and **b** NADPH. Increasing concentrations of pretomanid were incubated for 1.5 h at 30 °C, with $NAD^+$ and the other activating components, and either 100 μM or a titration of NADPH. The *Mt*-DprE1-DprE2 complex was incubated with the activated pretomanid for 10 min and DprE2 activity was measured after the addition of 200 μM or a titration of GGPR. Assays were performed in triplicate ($n = 3$) and mean initial rates (μM NADPH oxidized/min), with the standard error, were plotted using Prism GraphPad, fitting the curve using the model for uncompetitive inhibition (GGPR) and competitive inhibition (NADPH). (i) The fitted curve and (ii) double reciprocal plot. Background NADPH oxidation was subtracted. GGPR, geranylgeranylphosphoryl-D-ribose; $K_i$, inhibitor constant. Source data are provided as a Source Data file.

NADPH in two separate assays (Fig. 5). The curves were fitted with Prism GraphPad using the non-linear regression model for uncompetitive inhibition for GGPR and competitive inhibition for NADPH, both of which gave the best fit, correlating with their corresponding double reciprocal plots. The $K_i$ was calculated to be 4.9 ± 1.0 μM and 1.5 ± 0.4 μM for GGPR (alpha $K_i$) and NADPH respectively (± SEM).

Our initial attempts for these assays, whereby *Mt*-DprE1 or *Mt*-DprE1-DprE2 was incubated with pretomanid during activation for 3 h (Supplementary Fig. 6), showed that in addition to the inhibition of the DprE2 assay, a reduction in DprE1 activity for the *Mt*-DprE1-DprE2 complex was also observed ($IC_{50}$ of 25 ± 6.2 μM) (± SEM), though not for *Mt*-DprE1 alone. There was also a difference in the $K_i$ curves and reciprocal plots, which were fitted for non-competitive inhibition, with calculated $K_i$ values of 16.4 ± 1.9 μM and 14.1 ± 1.9 μM, for GGPR and NADPH respectively (± SEM). These higher values likely represent a deficiency of $NAD^+$ required for pretomanid activation, necessitating the additional assays.

## Discussion

We have confirmed that DprE2 is a target of two new clinically relevant *Mtb* drugs, delamanid and pretomanid, using whole cell target engagement studies and in vitro enzyme activity assays. Target identification has been problematic for these drugs, with spontaneous resistant mutant screens pointing to the Ddn-activation pathway and not the true target[10–12]. It was only when known inhibitors of DprE2

generated similar resistance profiles, that the possibility that these compounds all share a common target was realized[25]. DprE2, along with DprE1, form an essential epimerase complex in the synthesis pathway of DPA, a lipid-linked arabinose donor for the crucial *Mtb* cell wall component, arabinogalactan[18–21]. Previous studies that demonstrated reduced mycolic acids in drug-treated cells[6,7], were in fact close to the target, since a reduction in the arabinose content of the arabinogalactan domain likely decreases mycolic acid attachment sites. Further validation of DprE2 as the target is provided by metabolomic studies, which reported an increase in the levels of ribose-5-phosphate in pretomanid-treated cells[27]. Ribose-5-phosphate is a precursor in the synthesis pathway of DPA and could theoretically accumulate if the downstream pathway were blocked.

Pretomanid inhibition was demonstrated through both the DprE2 spectrophotometric and the radiolabeled assays. Although the $IC_{50}$ and $K_i$ values were reported here, the complex activation of the pretomanid pro-drug by Ddn into several different species means that the actual concentration of the active drug is not absolute. While there was no effect on DprE1 activity for either *Mt*-DprE1 or the *Mt*-DprE1-DprE2 complex when pre-activated pretomanid was added prior to the assay, the *Mt*-DprE1-DprE2 complex was inhibited to one-third of normal activity during assays that incubated the complex for 3 h during pretomanid activation. The method of pretomanid activation was also significant in the observed mode of inhibition calculated for the $K_i$ assays. After a short incubation time

with the activated pretomanid, DprE2 assays demonstrated that the active drug was competing with the NADPH for the active site, while the drug showed uncompetitive inhibition with GGPR. Following a longer incubation time during pretomanid activation, the inhibition was non-competitive for both substrates. This switch in the inhibition could indicate a slow change in the binding mode of the activated pretomanid from initial weak binding to tighter binding, a feature observed for the InhA inhibitor isoniazid, which is proposed to bind tightly once the final inhibited complex has formed[28]. Indeed the enzyme-inhibitor complex formed between activated pretomanid and DprE2 during a long incubation was found to be highly stable, withstanding a long period of dialysis that would serve to dilute a weakly bound compound. A more tightly bound pretomanid could also explain the inhibition of DprE1 activity observed after a long incubation with pretomanid. Tightly bound inhibitor could prevent substrate transfer from DprE1 to DprE2, thereby blocking *Mt*-DprE1 activity, a theory that is supported by the absence of DPX accumulation in our DPR epimerization studies.

We have endeavored to establish the active metabolite of pretomanid and delamanid and although not yet confirmed by mass spectrometry, our results strongly suggest it to be the previously discovered NAD-adduct[9,16]. This would not be the first instance of such an adduct, which has been reported for isoniazid and ethionamide during InhA inhibition[28–31]. Since DprE2 can utilize NADH/NADPH as cofactors, an NAD-pretomanid adduct would explain the initial competitive binding observed for NADPH and also the requirement for DprE2 when NAD(H) is not present, which provides a source of NAD(H) for adduct formation during activation. This finding also implies that NADH is the natural cofactor for DprE2, since it is confirmed here that NADPH cannot form an inhibitory adduct with pretomanid[9], only an endogenous supply of NAD(H) from the enzyme can form the active species. The separation of pretomanid activation and inhibition of DprE2 also indicates that the active form is not as short lived as initially thought[8]. Structural analysis of the pretomanid and delamanid pro-drugs, along with our recently discovered DprE2 inhibitors[25], show that they share a common nitro group (highlighted in Fig. 1a) attached to an aromatic ring. We suspect that it is this moiety that is reduced by Ddn during drug activation as previously suggested[9]. The alternative possibility, that pretomanid is reduced to a short-lived nitroso derivative that reacts directly with the enzyme, as observed for BTZ inhibition of DprE1[25,32], seems less likely with the results presented here. Though we also cannot discount that the multitude of other metabolites formed by Ddn activation of pretomanid and delamanid could be part of the multifaceted killing mechanism. These possibilities are all subject to future investigations.

Despite a concerted effort in the area, pretomanid and delamanid are some of the first new drugs to be licensed in over 40 years, specifically for patients with MDR and XDR strains of *Mtb*, those resistant to the current first- and second-line drugs. The discovery that a target of these two new antibiotics is DprE2, once more demonstrates the magnitude of the druggability of the DprE1-DprE2 complex. We envisage that this finding, together with accumulating evidence for the NAD-adduct as the active moiety, will accelerate discovery efforts into these clinically promising pro-drugs aiding further explorations into the development of new derivatives of the clinically successful pretomanid and delamanid.

## Methods
### Data collection and analysis
All data collected from the POLARstar Omega (BMG Labtech) plate reader were analyzed in Microsoft Excel 16.66.1 and GraphPad Prism 9.5.1.

### Mycobacterial strains and growth conditions
*M. bovis* BCG strain Pasteur was cultured in static liquid media (Middlebrook 7H9 (Difco), 10% (v/v) Middlebrook ADC, 0.25% (v/v) glycerol, 0.05% (v/v) Tween-80 at 37 °C, 5% CO$_2$. Media was supplemented with 25 µg/mL Kanamycin to select for mycobacterial plasmids.

For over-expression studies, mycobacteria were electroporated with mycobacterial expression constructs. Briefly, electrocompetent mycobacteria were prepared by pelleting and washing a mid-log culture of mycobacteria with decreasing volumes of room temperature 10% (v/v) glycerol. Cells were transferred to a 0.1 cm electrode gap electroporation cuvette, incubated with 1 µg plasmid DNA at 37 °C for 10 min, before a single pulse of 1.8 kV was applied. Cells were recovered overnight in liquid media before selection on solid medium containing 25 µg/mL Kanamycin.

### Preparation of plasmid expression constructs
The mycobacterial expression vector pMV261-*Mt-dprE1* was constructed using the primers (sense CTAGCTAGGGATCCAATGTTGAGCGTGGGAGCTAC, *Bam*HI; antisense CTAGCTAGAAGCTTCTACAGCAGCTCCAAGCGTCG, *Hind*III)[26], pMV261-*Mt-dprE2* was constructed with the primers (sense CTAGCTAGGGATCCAATGGTTCTTGATGCCGTAGG, *Bam*HI; antisense CTAGCTAGAAGCTTTCAGATGGGCAGCTTGCGGAAG, *Hind*III)[25] and pMV261-*Mt-inhA* was constructed with the primers (sense GATCGATCGGATCCAATGACAGGACTGCTGGACGG, *Bam*HI; antisense GATCGATCAAGCTTCTAGAGCAATTGGGTGTGCGC, *Hind*III). The *Escherichia coli* expression vector pCDF-duet-*Mt-dprE1* was constructed using primers (sense GATCGATCGGATCCGATGTTGAGCGTGGGAGCTACCACTACCG, *Bam*HI; antisense GATCGATCAAGCTTCTACAGCAGCTCCAAGCGTCGGGGCCATG, *Hind*III)[32] and the pCDF-duet-*Mt-dprE1-dprE2* vector was produced by additionally cloning the gene encoding *Mt*-DprE2 using the primers (sense CATGCATGCATATGGTCCTGGATGCTGTGGGC, *Nde*I; antisense CATGCATGCTCGAGTCAAATCGGCAGTTTACGG, *Xho*I)[25]. The gene encoding *Mt*-Ddn was cloned into the mycobacterial and *Escherichia coli* expression constructs pVV16 (sense primer, GATCGATCCATATGCCGAAATCACCGCCGCGGTTTC, *Nde*I; antisense primer, GATCGATCAAGCTTGGGTTCGCAAACCACGATCGGG, *Hind*III) and pET28SUMO (sense primer, GATCGATCGGATCCATGCCGAAATCACCGCCGCGGTTTC, *Bam*HI; antisense primer, GATCGATCAAGCTTTCAGGGTTCGCAAACCACGATCGGG, *Hind*III), respectively. The gene encoding *Mt*-Fgd1 was cloned into the *E. coli* expression construct pET28a (sense primer, GATCGATCCATATGGCTGAACTGAAGCTAGGTTACAAAGC, *Nde*I; antisense primer, GATCGATCAAGCTTTCAGCCAAGTCGCCGCAACC, *Hind*III). According to the manufacturer's instructions (Q5 High Fidelity DNA Polymerase, New England Biolabs), genes were amplified by PCR from *Mtb* H37Rv genomic DNA, purified (QIAquick PCR Purification Kit, Qiagen) and the PCR products and vectors were digested with the specified restriction enzymes (FastDigest, Thermo Fisher Scientific). The digest products were purified, ligated (T4 DNA ligase, New England Biolabs) and transformed into *E. coli* TOP10 cells. Positive clones were selected from solid medium (LB-agar with 50 µg/mL Kanamycin) and constructs were verified by DNA sequencing (Source Bioscience).

### Expression and purification of enzymes for in vitro assays
The recombinant proteins SUMO-Ddn and Fgd1 were over-expressed in *E. coli* BL21 (DE3) cells (from pET28SUMO and pET28a respectively). Cells were cultured in Terrific broth at 37 °C, with shaking, until OD$_{600nm}$ 0.4–0.6. Recombinant protein production was induced with 1 mM isopropyl-β-D-1-thiogalactopyranoside (IPTG) and cells were cultured overnight at 16 °C and subsequently harvested.

Recombinant proteins were purified through exploitation of the vector-encoded His-tag. Cell pellets were resuspended in binding buffer (50 mM sodium phosphate, pH 7.5, 0.5 M NaCl, 25 mM imidazole), and disrupted by sonication. Insoluble material was removed by centrifugation at 40,000 ×g, 4 °C, and the clarified lysate was applied to 1 mL nickel-charged immobilized metal affinity chromatography (IMAC) column (HisTrap IMAC HP, GE Healthcare) pre-equilibrated in

binding buffer. The column was washed and the recombinant protein eluted using a step gradient of imidazole in binding buffer (25, 50, 100, 150, 200, 1000 mM). Fractions containing purified recombinant proteins were dialyzed into 50 mM Tris, pH 7.5, 500 mM NaCl, 10% glycerol. To remove the His-tagged SUMO solubility tag from Ddn, SUMO protease was included with the purified recombinant SUMO-Ddn during dialysis. IMAC was repeated with binding buffer and a step gradient of imidazole (0, 10, 25, 50, 100, 300, 1000 mM) to separate Ddn from SUMO-Ddn and the His-tagged SUMO protease. Fractions containing purified Ddn were dialyzed into 50 mM Tris, pH 7.5, 500 mM NaCl, 10% glycerol. Recombinant Ddn and Fgd1 were concentrated and stored at −20 °C.

*Mt*-DprE1 alone and the *Mt*-DprE1-DprE2 complex were expressed from the pCDF-duet vector in *E. coli* BL21 (DE3) cells, along with the chaperones from the pTrc-*60.2-groES* plasmid[25,32,33]. Cells were grown at 37 °C, with shaking, to mid-log phase in terrific broth (Difco), supplemented with spectinomycin (100 μg/mL) and ampicillin (100 μg/mL), and then cooled to 20 °C in an ice bath. Expression was induced overnight at 20 °C with 0.5 mM IPTG, before pelleting cells. Cell pellets were resuspended in lysis buffer (50 mM Tris-HCL (pH 8), 50 mM NaCl and 10% glycerol), sonicated on ice, the insoluble material pelleted at 75,000 ×g, 4 °C and the soluble material passed through a HisTrap column. Proteins were washed in lysis buffer with 20 mM imidazole and eluted with 300 mM imidazole. A further purification step was performed with a QHP column in lysis buffer with increasing NaCl concentrations. The *Mt*-DprE1-DprE2 complex passes straight through the column, whilst *Mt*-DprE1 alone elutes at 120-140 mM NaCl. Proteins were dialyzed in lysis buffer with 200 mM NaCl, concentrated and stored at −80 °C.

## MIC determination

The liquid MIC of mycobacterial over-expressor strains (*M. bovis* BCG harboring the expression constructs pMV261, pMV261-*Mt-dprE1*, pMV261-*Mt-dprE2* and pMV261-*Mt-inhA*) with different compounds was evaluated using the microplate alamar blue assay (MABA). Briefly, mid-log cells (OD$_{600nm}$ 0.4–0.8) were diluted to $1 \times 10^6$ colony forming units (CFU)/mL (OD$_{600nm}$ of 1, equates to $2.5 \times 10^8$ CFU/mL) in the specified liquid growth medium. 99 μL cells were added to individual wells of a 96-well plate (black walls, black, flat bottom; Greiner) along with 1 μL 100 × desired concentration of compound (increasing concentrations across the plate). All compounds were dissolved in 100% DMSO, with the exception of isoniazid, which was dissolved in water. Plates were incubated at 37 °C, 5% CO$_2$ for 7 days. 30 μL 0.02% (w/v) resazurin and 12.5 μL 20% Tween 80 was added to each well and the plates were incubated for a further 24 h. Cell viability was established by measuring fluorescence (λ excitation 530 nm, λ emission 590 nm) using a POLARstar Omega plate reader. Data were fitted using nonlinear regression (Prism, GraphPad).

## CFU enumeration

Mid-log (OD$_{600nm}$ 0.4–0.8) *M. bovis* BCG harboring the expression constructs pMV261, pMV261-*Mt-dprE1* or pMV261-*Mt-dprE2* were diluted to $1 \times 10^6$ colony forming units (CFU)/mL. In a 96 well plate, 99 μL cells were treated in triplicate with the desired drug concentration of pretomanid, delamanid, BTZ or INH and incubated at 37 °C, 5% CO$_2$ for 7 days. Cells were resuspended and 5 μL of each suspension was directed spotted on to 7H11 agar containing 25 μg/mL Kanamycin. A further 5 μL was serially diluted from $10^{-1}$ to $10^{-6}$. Before spotting on to solid media. The agar plates were incubated for 2 weeks at 37 °C before CFUs were counted.

## Analysis of the effect of pretomanid and delamanid on arabinogalactan biosynthesis

The effect of pretomanid and delamanid on whole cell arabinogalactan biosynthesis was evaluated using [$^{14}$C]-labeling. The study was performed with *Mycobacterium smegmatis* mc$^2$155 owing to the inability of *M. bovis* BCG to uptake the required radiolabel ([$^{14}$C]-glucose). Wild-type *M. smegmatis* is inherently resistant to pretomanid and delamanid and so *M. smegmatis* harboring the mycobacterial expression construct pVV16-*ddn* was used in the labeling experiments: constitutive Ddn over-expression enabled pro-drug activation, reducing the MICs of delamanid from >10 μM to 0.5 μM and pretomanid from >200 μM to 2.5 μM.

*M. smegmatis* strain mc$^2$155 was cultured in static or shaking liquid Luria-Burtani (LB) broth with 0.05% (v/v) Tween-80 or Glycerol-Alanine-Salts (GAS) media (Bacto casitone, 100 μM ferric ammonium citrate, 23 mM K$_2$HPO$_4$, 10 mM citric acid, 11 mM L-alanine, 6 mM MgCl$_2$−6H$_2$O, 3 mM K$_2$SO$_4$, 37 mM NH$_4$Cl, 1% (v/v) glycerol, pH 6.6) with 0.05% (v/v) Tween-80 at 37 °C. For growth on solid medium, LB-agar was used. Media was supplemented with 25 μg/mL Kanamycin to select for the mycobacterial plasmid, pVV16-*ddn*.

The pVV16-*ddn* mycobacterial expression construct was generated using standard cloning procedures (sense primer, GATCGATCC ATATGCCGAAATCACCGCCGCGGTTTC, *Nde*I; antisense primer, GATCGATCAAGCTTGGGTTCGCAAACCACGATCGGG, *Hind*III) and electroporated into *M. smegmatis*.

*M. smegmatis* mc$^2$155 harboring the vector pVV16-*ddn* was cultured to mid-log (OD$_{600nm}$ 0.4–0.8) and diluted to OD$_{600nm}$ 0.2. 10 mL cell cultures were treated with a dose-dependent increase in drug (0x, 0.5x and 1x MIC; MICs: delamanid, 0.5 μM; pretomanid, 2.5 μM; BTZ, 15 nM) for 6 h before adding 1μCi/mL [$^{14}$C]-D-glucose (PerkinElmer) for a further 16 h. Cells were harvested, and pellets were washed in phosphate buffered saline. Cells were extracted 3 times in 3 mL 2:1 chloroform:methanol at 50 °C for 2 h. The pellet was treated 3 times with 1 mL 50% (v/v) ethanol, 80 °C for 1 h, followed by 3 times 1 mL 2% (w/v) SDS, 90 °C for 1 h. The pellet was then resuspended in 1 mL 80% (v/v) acetone, solvent discarded, and resuspended in 1 mL 100% (v/v) acetone. The solvent was discarded and the pellet was then dried. The arabinogalactan pellet was hydrolyzed for 3 h in 300 μL 2 M trifluoroacetic acid at 120 °C. The trifluoroacetic acid was evaporated and the resulting sugars were analyzed by TLC (aluminum-backed HPTLC silica gel 60 F$_{254}$) in 15:1 (v/v) acetone:water, using equal counts (50, 000 cpm) and exposed to Kodak X-Omat film for 5 days. Sugars were identified using cold standards. The relative abundance of arabinose and galactose were determined by densitometry.

## [$^{14}$C]-labeled-DPR radioactive assays

Pretomanid inhibition of DprE2 in the presence of DprE1 was addressed by monitoring the conversion of [$^{14}$C]-DPR to [$^{14}$C]-DPA. [$^{14}$C]-DPR was prepared as previously[34]: 100 μCi (100 mCi/mmol) of dried [$^{14}$C]-glucose (PerkinElmer) was resuspended in 800 μL buffer (50 mM Hepes pH 7.6, 2 mM MgCl$_2$ and 0.5 mM MnCl$_2$), adding 10 units of hexokinase, 2 mM ATP, 8 mM NADP. After a 2 minute incubation at room temperature, 10 units of glucose-6-phosphate dehydrogenase, 2 units of 6-phosphogluconate dehydrogenase, 10 units of phosphoriboisomerase and 1 unit of phosphoribosylpyrophosphate synthetase was added and the mixture incubated at 37 °C for 30 minutes. A further 3 mM ATP was added slowly over an hour. The resulting p[$^{14}$C]Rpp was filtered through a 1.5 mL 10 kDa centrifugal filter (Amicon), to remove the enzymes, and separated from intermediate reaction products and substrates using a 3 mL LC-SAX SPE column (Merck), washing with water and eluting with a gradient of sodium acetate in water (300 mM to 2 M). The fractions were visualized on a glass backed cellulose TLC, separated in 0.85 M phosphate buffer pH altered to 3.4 with phosphoric acid and exposed to a Kodak X-Omat film for 1 night. 1 million counts of p[$^{14}$C]Rpp was converted to [$^{14}$C]-DPR by incubating with 100 μL BTZ inhibited *M. smegmatis* membranes (100 μM BTZ in a total of 300 μL of 50 mM MOPs buffer pH 7.9, 1 mM ATP, preincubated for 10 minutes at 37 °C) and 15 μL decaprenyl phosphate (C50-P) (1 mM in 1% CHAPs) for 1.5 h at 37 °C. [$^{14}$C]-DPR was separated by adding 3.7 mL

of 10:10:3 ($CHCl_3$: $CH_3OH$: $H_2O$) (v/v) and incubating at room temperature for 8 h. Then 750 μL $H_2O$ and 1.75 mL $CHCl_3$ was added to set up a biphase, washing the lower organic layer twice with 3:47:48 ($CHCl_3$: $CH_3OH$: $H_2O$) (v/v), drying the lower organic fraction.

Increasing concentrations of pretomanid (0, 50, 100, 200 μM) were activated overnight at room temperature in buffer (50 mM sodium phosphate, pH 7.5, 100 mM NaCl) with 4 μM Fgd1, 36 μM $F_{420}$, 1 mM glucose-6-phosphate, 5 μM Ddn, 300 μM NADPH, 10 μM $Mt$-DprE1-DprE2. Control reactions were also prepared: (lane 1) reaction buffer; (lane 2) reaction buffer with 10 μM $Mt$-DprE1, 300 μM NADPH; (lane 3) reaction buffer with 10 μM $Mt$-DprE1-DprE2, 300 μM NADPH; (lane 8) 200 μM pretomanid in the absence of Ddn. The reactions were initiated with 5 μL GGPR and 5.5 μL of [$^{14}$C]-DPR (2,000 cpm) in 1% IGEPAL for 1 h at 37 °C and then quenched with 350 μL of $CHCl_3$:$CH_3OH$ (2:1 (v/v)). The lower layer of the biphase was dried, resuspended in 10 μL of $CHCl_3$:$CH_3OH$ (2:1) (v/v), spotted onto a high performance silica TLC (Merck) and separated by $CHCl_3$:$CH_3OH$:$NH_4OH$:$H_2O$ (65:25:0.5:3.6, (v/v/v)). Bands were visualized following exposure to Kodak X-Omat film for 2 weeks.

### Spectrophotometric assays for DprE1 and DprE2 activity

Pretomanid was activated for 1.5 h at 30 °C in the presence of NAD$^+$. The following reagents were added to a final volume of 22 μL: reaction buffer (50 mM sodium phosphate, pH 7.5, 4 mM $MgCl_2$, 100 mM NaCl), 5 μM Fgd1, 12 μM $F_{420}$, 1 mM glucose-6-phosphate and 200 μM NAD$^+$. 1 μL of serially diluted pretomanid in DMSO was added with increasing concentrations in a dose response format (0, 0.78, 1.56, 3.125, 6.25, 12.5, 25, 50 μM). 5 μM Ddn initiated the activation of the pro-drug. Equivalent control reactions were prepared in the absence of Ddn. The active pretomanid drug was then incubated for 10 min at room temperature with either 10 μM $Mt$-DprE1 alone or 10 μM $Mt$-DprE1-DprE2 complex. Next 100 μM NADPH was added and the DprE1[26] and DprE2[25] assays were performed, aliquoting the assay mix (23 μL) into Greiner 384 black-bottom plates. For the DprE1 assay, 100 μM resazurin (2 mM stock) was added to the reactions before they were loaded onto the plate. Reactions were measured (DprE2 assay: λ excitation 340 nm, λ emission 445 nm; DprE1 assay: λ excitation 530 nm, λ emission 590 nm), 37 °C, using a POLARstar Omega (BMG Labtech) plate reader. The reaction was initiated by the addition of 200 μM of GGPR (2.5 mM stock) using a multichannel pipette. Reactions were performed in triplicate.

Assays to determine the $K_i$ with varying substrate concentrations were performed as above, using increasing concentrations of pretomanid (0, 5, 10, 25, 50, 75, 100 μM), incubating with 200 μM NAD$^+$ for 1.5 h at 30 °C, with the activating enzymes and constituents. Following a 10 min incubation at room temperature of the active pretomanid with the $Mt$-DprE1-DprE2 complex, DprE2 activity was measured after the addition of 100 μM NADPH and a titration of GGPR (0, 25, 50, 75, 100, 150, 200 μM) or 200 μM GGPR and a titration of NADPH (6.125, 12.5, 25, 50, 100, 200 μM) (λ excitation 340 nm, λ emission 445 nm) at 37 °C, using a POLARstar Omega (BMG Labtech) plate reader. Reactions were performed in triplicate.

For the activation requirement assay, 200 μM pretomanid was incubated with the activating enzymes and constituents for 3 h at 30 °C, with either 100 μM NADH, NADPH, NAD$^+$, NADP$^+$ or no nicotinamide cofactor, plus/minus 10 μM $Mt$-DprE1-DprE2 complex and plus/minus 10 μM Ddn. Following pretomanid activation, 10 μM $Mt$-DprE1-DprE2 complex was incubated for 10 min at room temperature with the assays that lacked the complex. 100 μM NADPH was added to every assay (except where already present) and DprE2 activity was measured after the addition of 200 μM of GGPR (2.5 mM stock). Reactions were performed in triplicate.

During initial assays, the requirement appeared to be for DprE2 and NAD(H) was not known to be necessary. Therefore, pretomanid was activated in the presence of the $Mt$-DprE1-DprE2 complex (or $Mt$-DprE1 alone) for 3 h at 30 °C in the presence of 100 μM NADPH and the other components for activation (50 mM sodium phosphate, pH 7.5, 4 mM $MgCl_2$, 100 mM NaCl), 5 μM Fgd1, 5 μM Ddn, 12 μM $F_{420}$, 1 mM (glucose-6-phosphate) in a final volume of 22 μL. $IC_{50}$ values were measured by adding 1 μL of increasing concentrations of pretomanid. $K_i$ assays were also performed by activating a titration of pretomanid in this way, titrating either NADPH during the activation period or GGPR to initiate the reaction, keeping the other substrate constant. DprE1 and DprE2 activity were measured as normal.

For the dialysis experiment, the assay mixture was increased to 1 mL keeping the concentrations of all assay components constant. The $Mt$-DprE1-DprE2 complex was inhibited with 400 μM pretomanid, plus and minus the Ddn activator. After a 3 h incubation at 30 °C, half of each assay was dialyzed overnight into 50 mM potassium phosphate, 500 mM KCl, 10% glycerol at 4 °C, using 14 kDa cut-off dialysis tubing (Sigma). The other half of the mixture was stored at 4 °C. The $OD_{280nm}$ was measured for each sample using a nanodrop and the concentration of the pre-dialysis samples was reduced using the dialysis buffer to account for the diluting effect of dialysis. 20 μL of the samples, pre- and post-dialysis, were loaded into Greiner 384 black-bottom plates, spiking the dialyzed sample with 100 μM NADPH. The reaction was initiated with 200 μM of GGPR (1 mM stock), and DprE2 activity measured (λ excitation 340 nm, λ emission 445 nm) at 37 °C, using a POLARstar Omega (BMG Labtech) plate reader. Reactions were performed in triplicate.

### Reporting summary
Further information on research design is available in the Nature Portfolio Reporting Summary linked to this article.

## Data availability
The data generated in this study (Fig. 2, fluorescence counts and CFUs; Fig. 3, uncropped TLC; Figs. 4 and 5, fluorescence counts; Supplementary Fig. 1, fluorescence counts and CFUs; Supplementary Fig. 2, densitometry values; Supplementary Figs. 3, 4, 5 and 6, fluorescence counts) are provided with this paper in the Source Data file. Source data are provided with this paper.

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

## Acknowledgements
The authors thank Albel Singh for photographing TLCs for the figures. Funding has been awarded as follows: Personal Research Chair from Mr. James Bardrick, G.S.B. Medical Research Council MR/S000542/1, G.S.B. Medical Research Council MR/R001154/1, G.S.B. Sir Charles Hercus Health Research Fellowship awarded through the Health Research Council of New Zealand, G.B.

## Author contributions
Conceptualization: K.A.A., S.M.B., G.S.B. Methodology: K.A.A., S.M.B., G.S.B. Investigation: K.A.A., S.M.B., S.S., N.V., G.B., G.S.B. Visualization: K.A.A., S.M.B., G.S.B. Supervision: G.S.B. Writing—original draft: K.A.A., S.M.B., G.S.B. Writing—review and editing: K.A.A., S.M.B., G.S.B.

## Competing interests
The authors declare no competing interests.
