## [Peer Review File · Nature Communications]

DprE2 is a molecular target of the anti-tubercular nitroimidazole compounds, pretomanid and delamanidReviewer #1 (Remarks to the Author):

Abrahams et al identified the decaprenylphosphoribose-2'-epimerase DprE2 as a target of the activated nitroimidazoles pretomanid and delamanid. They demonstrate that ectopic expression of DprE2 results in increased resistance to both pretomanid and delamanid, while DprE1 expression specifically shifted the MIC of its known inhibitor BTZ. They further performed in vitro assays that confirmed that activated pretomanid inhibits DprE2, while the prodrug did not. Interestingly in the presence of both complex members, DprE1 activity was inhibited as well, although the mechanism by which this occurs remains unclear. Finally, they provide evidence that the activation of pretomanid requires DprE2 presence suggesting that activated pretomanid is unstable/short-lived in the absence of DprE2.

Overall, this is a conclusive study establishing DprE2 and thus arabinogalactan biosynthesis as a target of pretomanid and delamanid. While the activated form of pretomanid remains unidentified, the target identification of pretomanid and linezolid will be of interest to the field and allow development of additional DprE2 inhibitors. The authors should acknowledge and cite a recent paper that demonstrated that pretomanid forms an NIH adduct (Kreutzfeldt et al. Nat Comm 2021; PMID: 35459278).

Reviewer #2 (Remarks to the Author):

This paper reports the identification of the DprE2 enzyme as the cellular target of the Ddn activated nitroimidazole metabolites. The work is the continuation of the previous research of the authors (reported in the paper in press, attached during submission), in which the GSK-TBset library was screened against a DprE2 overexpressing *M. bovis* strain. The isolation and whole genome sequencing of resistant mutants against some hit compounds emerged from this screening revealed mutation in genes *fgd1* and *fbiC*, found also in pretomanid and delamanid resistant isolates, suggesting that these compounds share the same mechanism of activation and possibly the same target(s). Indeed, further experiments presented in the current work, using both DprE2 overexpressing mycobacterial cells and recombinant DprE1-DprE2 enzyme complex, demonstrated that pretomanid and delamanid active metabolites are actually specific inhibitors of the reductase. Unfortunately, it was not possible to isolate and characterise the precise active metabolites, probably for its high instability.

The results are of particular importance since despite nitroimidazoles have been approved for TB treatment, their precise mechanism of action is currently mainly still unknown, particularly regarding the target(s) of the activated metabolites. Moreover, the identified target DprE2 is of particular interest, as it has been reported to work in concert with DprE1, a well-known promiscuous drug target, thus postulated as a potential drug target, although no active compounds have been reported until now.

Overall, the work is very well designed and conducted, using the appropriate methodology. The results are clearly described, and the discussion well-articulated and punctual. Thus I do not have particular criticisms.

Reviewer #3 (Remarks to the Author):

Despite the excellent in vivo efficacies against both drug-sensitive and drug-resistant *Mtb*, the two recently approved bicyclic nitroimidazoles, Pretomanid and Delamanid, remain largely enigmatic pertaining to their mechanisms of action in killing *Mtb*. In this draft, Abrahams et al. report biochemical and cellular evidence suggesting that the essential decaprenylphosphoribose-2-epimerase, DprE2, could be the intracellular target of the activated forms of Pretomanid and Delamanid. In a separate study, the authors characterized several nitro compounds that were active against *Mtb* and evidently targeted DprE2, and mapped their resistance mutations to *fgd1* and *fbiC*, both of which are critical for the activation of Pretomanid or Delamanid. While these compounds are structurally distinct from bicyclic nitroimidazoles apart from their F420 reactive nitro groups, the authors hypothesized that Pretomanid and Delamanid might also inhibit DprE2 with a similar mechanism. They showed that ectopic overexpression of DprE2, but not DprE1,

conferred resistance to Pretomanid/Delamanid, and provided biochemical data whereby co-incubation of a pretomanid-activating reaction mix with DprE1-E2 for a prolonged period led to inhibition of DprE2's epimerase activity as well as its ability to oxidize NAD(P)H.

Overall, the manuscript is well-written and easy to follow. The enzymatic assays are particularly interesting although most of the assays were established for publication in a separate manuscript. The hypothesis is genuinely interesting and tempting, as it does track with orthogonal evidence whereby pretomanid treated Mtb phenocopied those that were challenged by other cell wall-targeting compounds (PMID: 32680963). However, I'm afraid that this manuscript lacks cellular evidence that corroborates this hypothesis and there might be issues with the interpretation of the biochemical data as well. Specifically, my concerns are:

Major issues/suggestions:

1. The cellular evidence of DprE2 being a target of pretomanid/delamanid is generally weak. As DprE2 uses NAD(P)H as the cofactor(s), an alternative explanation for the observed desensitization by DprE2 overexpression (especially when expressed from the multi-copy, episomal plasmid pMV261) would be that DprE2 might act as a natural sink for the toxic, pretomanid/delamanid derived NAD-adducts (Hayashi et al., 2020), which is fundamentally different from DprE2 being the primary/secondary target of pretomanid/delamanid in vivo. If DprE2 is truly an important target of pretomanid/delamanid, one would expect to see: decreased LAM/LM ratio in crude or purified cell envelope and diminished arabinose abundance in hydrolyzed cell wall sugars (PMID: 7793875, 24446451); increased cell envelope permeability (PMID: 32974492, which indeed confirmed that pretomanid facilitates rifampicin penetration); compromised cell envelope structure by electron or fluorescence microscopy. I noticed that the senior authors of this draft published an elegant study back in 2014 which clearly demonstrated that DprE1 inhibitor, BTZ043, led to the depletion of the cell wall LAM/arabinose (PMID: 24446451, Fig. 6&7). The authors should consider leveraging these existing platforms to demonstrate direct changes of the cell envelope upon treatment with pretomanid/delamanid.
2. The authors used only alamar blue to evaluate cell viability after drug treatment. First of all, alamar blue is probably not a great measure of "viability" per se, as its readout is confounded by cellular growth, redox homeostasis, and the metabolic state of the cells. Secondly, it is known that bicyclic nitroimidazoles, especially when used at high concentrations, could be converted to various reactive species which may perturb intracellular redox balance and obscure the interpretation of alamar blue data. CFU enumeration could address these complications (PMID: 24446451, Fig. 4C).
3. I noticed that the effective concentrations of pretomanid used for the in vitro biochemical assays are hundreds of folds higher than the in vivo MICs. While it is understandable that the reconstituted Fgd1-F420-Ddn reaction is potentially less efficient as compared to the endogenous system, the authors did not provide any data on the kinetics of pretomanid activation and the subsequent deactivation of the reactive derivatives. To justify the necessity of co-incubating pretomanid activation reaction with DprE1E2 reaction for a prolonged period, the authors should measure the 1). the proportion of pretomanid that is converted by Ddn and 2). the relative abundance and half-lives of des-nitro products and other by products by mass spectrometry. Also, I would suggest using K_i instead of IC_{50} to denote inhibition efficiency in Fig. 4, as IC_{50} is affected by substrate concentration, mode of action etc., and doesn't quite facilitate the intuitive comparison between cell-free biochemical and whole-cell data.
4. One straightforward experiment to test the NAD adduct hypothesis is to assess whether synthesized delamanid-NAD adduct (Hayashi et al., 2020) is sufficient to inhibit DprE2 enzymatic activity, which does not rely on the in vitro activation of pretomanid/delamanid.
5. The reactive nitrogen species (de-nitro derivatives, nitric oxide, etc.) released upon pretomanid activation may directly damage the enzymes and lead to irreversible inactivation. One way to demystify this is to dialyze the 3hour/overnight Pretomanid-DprE1E2 mixture to remove residual pretomanid and its derivatives and all cofactors, then measure DprE1 and DprE2 enzymatic activities after supplementing FAD, GGPR, +/- NADPH. If DprE2 enzymatic activity is restored after dialysis, it suggests that the inhibition is reversible and that the inhibitory pretomanid derivative probably has a relatively high dissociation constant. If not, it implies that either the reactive derivative(s) altered the chemical composition or folding/aggregation of the DprE2 enzyme, or that the inhibitory derivative binds extremely tight.

Minor issues:

1. The authors may consider providing the unprocessed time-fluorescence curves for DprE2-mediated NADPH oxidization in response to increasing concentrations of pretomanid.
2. Ln. 320-322: figure reference would be helpful.
3. Ln. 477-482: I believe that FAD⁺ is also required for the initiation of the reaction, which is missing.
4. Ln. 261: "...as the intermediate GGPR is reduced to the GGPR product...". GGPR product should be GGPA?

Response to Reviewers' Comments

We are grateful for the reviewers' comments and are pleased to revise the manuscript based on the reviewers' suggestions.

Reviewer #1

The authors should acknowledge and cite a recent paper that demonstrated that pretomanid forms an NIH adduct (Kreutzfeldt et al. Nat Comm 2021; PMID: 35459278).

Response: This paper has been acknowledged and we now provide further evidence for the NAD-adduct as the active form of pretomanid.

Reviewer #2

No suggestions made.

Response: We are pleased that the manuscript satisfies **Reviewer #2** and hope that our revisions further improve our manuscript.

Reviewer #3

1. The cellular evidence of DprE2 being a target of pretomanid/delamanid is generally weak. As DprE2 uses NAD(P)H as the cofactor(s), an alternative explanation for the observed desensitization by DprE2 overexpression (especially when expressed from the multi-copy, episomal plasmid pMV261) would be that DprE2 might act as a natural sink for the toxic, pretomanid/delamanid derived NAD-adducts (Hayashi et al., 2020), which is fundamentally different from DprE2 being the primary/secondary target of pretomanid/delamanid in vivo. If DprE2 is truly an important target of pretomanid/delamanid, one would expect to see: decreased LAM/LM ratio in crude or purified cell envelope and diminished arabinose abundance in hydrolyzed cell wall sugars (PMID: 7793875, 24446451); increased cell envelope permeability (PMID: 32974492, which indeed confirmed that pretomanid facilitates rifampicin penetration); compromised cell envelope structure by electron or fluorescence microscopy. I noticed that the senior authors of this draft published an elegant study back in 2014 which clearly demonstrated that DprE1 inhibitor, BTZ043, led to the depletion of the cell wall LAM/arabinose (PMID: 24446451, Fig. 6&7). The authors should consider leveraging these existing platforms to demonstrate direct changes of the cell envelope upon treatment with pretomanid/delamanid.

Response: We recognize the concern of **Reviewer #3** with regards to the need for whole cell target engagement of pretomanid and delamanid. Therefore, we conducted a [¹⁴C]-whole cell labeling experiment in the presence of increasing concentrations of pretomanid, delamanid and BTZ (positive control), extracted the arabinogalactan and analyzed the relative percentages of arabinose and galactose. The results show that the relative percentage of arabinose decreased, whilst galactose increased with increasing concentrations of each drug (Supplementary Figure 2). We hope this data provides sufficient evidence confirming the inhibitory impact of pretomanid and delamanid on the synthesis of the cell wall envelope.

2. The authors used only alamar blue to evaluate cell viability after drug treatment. First of all, alamar blue is probably not a great measure of “viability” per se, as its readout is confounded by cellular growth, redox homeostasis, and the metabolic state of the cells. Secondly, it is known that bicyclic nitroimidazoles, especially when used at high concentrations, could be converted to various reactive species which may perturb intracellular redox balance and obscure the interpretation of alamar blue data. CFU enumeration could address these complications (PMID: 24446451, Fig. 4C).

Response: CFU enumeration has been completed (new Figure 2) to alleviate concerns of viability determined from MABA.

3. I noticed that the effective concentrations of pretomanid used for the in vitro biochemical assays are hundreds of folds higher than the in vivo MICs. While it is understandable that the reconstituted Fgd1-F420-Ddn reaction is potentially less efficient as compared to the endogenous system, the authors did not provide any data on the kinetics of pretomanid activation and the subsequent deactivation of the reactive derivatives. To justify the necessity of co-incubating pretomanid activation reaction with DprE1E2 reaction for a prolonged period, the authors should measure the 1). the proportion of pretomanid that is converted by Ddn and 2). the relative abundance and half-lives of des-nitro products and other by-products by mass spectrometry. Also, I would suggest using K_i instead of IC_{50} to denote inhibition efficiency in Fig. 4, as IC_{50} is affected by substrate concentration, mode of action etc., and doesn't quite facilitate the intuitive comparison between cell-free biochemical and whole-cell data.

Response: Based on the comments from **Reviewer #3**, we have re-investigated the parameters of pretomanid activation and subsequent inhibition. We too, were perplexed by the prolonged incubation time required to activate pretomanid and inhibit DprE2. Given the NAD-adduct observed from extracts of pretomanid and delamanid-treated *Mtb*, we analyzed DprE2 inhibition of pretomanid activated in the presence of the co-factors NAD^+ , NADH, $NADP^+$ and NADPH, with and without the *Mt*-DprE1-DprE2 complex present during pro-drug activation. We discovered that DprE2 does not need to be present, as initially believed, and that DprE2 is inhibited by a metabolite that is generated when pretomanid is activated in the presence of NAD(H) and not NADP(H). All of our initial studies were performed with NADPH due to the interference of NADH oxidation in the presence of Fgd1 (DprE2 can use NADH or NADPH as a co-factor). Because the active metabolite of pretomanid is not formed in the presence of NADPH, this provides an explanation for the long incubation of DprE2 with the activating components required to observe inhibition: the active metabolite was likely generated with NAD(H) that had co-purified with DprE1-DprE2 complex. This suggests that NADH is the natural co-factor of DprE2. We no longer require a long incubation time of DprE2 with activated pretomanid to observe inhibition.

We are actively pursuing the identification of the active metabolite through mass spectrometry. Despite efforts, we have been unable to establish the proportion of pretomanid converted by Ddn to the active metabolite, because 1) there are numerous break-down products and 2) the active metabolite may be unstable and/or form other

inhibitory derivatives with components of the assays (co-factors for example). We are aware that the proportion of pretomanid converted does not necessarily equate to the amount of active metabolite available. We are in the process of optimizing mass spectrometry conditions to analyze inhibitor-enzyme complexes and solvent extractions of the inhibited DprE2 complex. Our preliminary data has not been able to identify a mass equivalent to the NAD-adduct, due to significant background signals. We are investigating LC-MS/MS, as performed in PMID: 35459278, to identify the presence of the NAD-adduct from dialyzed inhibited DprE2 reactions and this is subject to ongoing investigations.

As recommended, the K_i values with respect to NADPH and GGPR have been calculated. The data suggests that the inhibition is uncompetitive with respect to GGPR and competitive with respect to NADPH. Interestingly, when pretomanid was activated in the presence of DprE2 with a long incubation and then dialyzed (see point 5), DprE2 remained inhibited, suggesting irreversible inhibition. We imagine a scenario similar to INH-NAD-adduct inhibition of InhA, whereby the initial enzyme-inhibitor complex is formed rapidly, followed by the conversion to a final inhibited complex, with a slow rate of adduct dissociation.

4. One straightforward experiment to test the NAD adduct hypothesis is to assess whether synthesized delamanid-NAD adduct (Hayashi et al., 2020) is sufficient to inhibit DprE2 enzymatic activity, which does not rely on the in vitro activation of pretomanid/delamanid.

Response: We strongly agree with **Reviewer #3** that this would be an advantageous experiment to perform. Unfortunately and disappointingly, it is beyond our capabilities to chemically synthesize the delamanid-NAD or pretomanid-NAD adduct and the methods to synthesize the delamanid-NAD adduct are not reported in Hayashi et al., 2020. However, now we have further evidence that the NAD-adduct is the active metabolite, we are investigating whether we can purify the NAD-adduct following enzyme activation. We also hope that future mass spectrometry of the dialyzed inhibited *Mt*-DprE1-DprE2 complex will confirm the active metabolite.

5. The reactive nitrogen species (de-nitro derivatives, nitric oxide, etc.) released upon pretomanid activation may directly damage the enzymes and lead to irreversible inactivation. One way to demystify this is to dialyze the 3hour/overnight Pretomanid-DprE1E2 mixture to remove residual pretomanid and its derivatives and all cofactors, then measure DprE1 and DprE2 enzymatic activities after supplementing FAD, GGPR, +/- NADPH. If DprE2 enzymatic activity is restored after dialysis, it suggests that the inhibition is reversible and that the inhibitory pretomanid derivative probably has a relatively high dissociation constant. If not, it implies that either the reactive derivative(s) altered the chemical composition or folding/aggregation of the DprE2 enzyme, or that the inhibitory derivative binds extremely tight.

Response: We have performed the dialysis of the *Mt*-DprE1-DprE2 complex with activated and non-activated pretomanid and analyzed *Mt*-DprE2 activity post dialysis. The activity was compared to equivalent reactions that were not dialyzed. The activity of inhibited and uninhibited *Mt*-DprE2 of dialyzed and non-dialyzed reactions are similar: *Mt*-DprE2 with activated pretomanid remains inhibited after dialysis. This implies that the inhibitory derivative binds tightly, or even covalently. We do

not believe the over-night dialysis impacted the integrity of the enzyme, given the activity of *Mt*-DprE2 with non-activated pretomanid.

In Figure 4 we addressed the cofactor requirements of pretomanid activation for the inhibition of *Mt*-DprE2. If the reactive nitrogen species upon pretomanid activation directly damaged the enzymes causing irreversible inactivation, then we would not expect to see any differences of inhibition of DprE2 activity upon activation of pretomanid in the presence of different cofactors. During co-incubation of DprE2 with the activating components of pretomanid, inhibition is observed with each cofactor. However, when pretomanid is activated prior to incubation with DprE2, inhibition is only detected with NAD⁺ or NADH. This suggests specificity of the active metabolite, and not generic damage caused by reactive nitrogen species. In the original Figure 4Cii, with the long incubation times with the pro-drug activation components, DprE1 activity alone was not inhibited suggesting that the reactive nitrogen species were not directly damaging the enzymes, or if they were, they had specificity for DprE2 alone.

Reviewer #3 (Remarks to the Author):

The revised manuscript by Abrahams et al has addressed many of the major concerns I had. Specifically, the authors have conducted additional enzymatic assays to investigate the chemical nature of the active intermediate(s). They have also provided preliminary evidence indicating that pretomanid activation generates NADP-adduct(s) that specifically inhibits DprE2. Although it is still unclear whether DprE2 inhibition is the dominant MOA of pretomanid and delamanid, this study has clearly demonstrated that DprE1E2 is an important yet missing piece of the puzzle. I believe this will inform future studies in demystifying the bactericidal activities of these compounds and designing new structural scaffolds that target this critical biosynthetic pathway. I congratulate the authors on an excellent manuscript that should be of broad interest to the general audience of Nature Communications.